# The Effects of Factors of Production Shocks on Labor Productivity: New Evidence Using Panel VAR Analysis

**Nurliyana Mohd Basri**, **Zulkefly Abdul Karim \*** and **Noorasiah Sulaiman**

Center for Sustainable and Inclusive Development Studies (SID), Universiti Kebangsaan Malaysia (UKM), Bangi, Selangor 43600, Malaysia; nurliyana.phd@gmail.com (N.M.B.); rasiahs@ukm.edu.my (N.S.)

\* Correspondence: zak1972@ukm.edu.my

**Abstract:** Labor productivity has an essential role in creating a more sustainable labor market platform, leading to better economic sustainability. However, the sluggish growth in labor productivity in Malaysia could hinder the vision in realizing the status of a high-income nation in the future. Thus, understanding how production shocks affect labor productivity sustainability is crucial for firms in managing their inputs (resources). This paper aims to elucidate how shocks in wage, capital intensity, and human capital may affect the dynamic of labor productivity in the Malaysian manufacturing industry. The study further explains the magnitude of this impact on labor productivity. This study employs the panel vector autoregression (PVAR) model in analyzing the propagation of the shocks through the impulse response function and variance decomposition. The main findings reveal that shocks in production factors have a positive and significant transitional impact on productivity and the cumulative effects are positive over time. The economic impact of wage shock is material, whereas capital intensity shock is moderate and only exerts a minor effect on labor productivity emanating from human capital shock. These findings provide further insights into assisting policymakers in amplifying the current labor market policy for sustainable economic growth.

**Keywords:** production shocks; labor productivity sustainability; panel vector autoregression (PVAR); manufacturing industry; Malaysia

## 1. Introduction

Sustainability accounts for the perseverance of productive capacity for the indefinite future with the socio-economic aspect of human aspirations for better welfare, well-being, and development [1]. Labor productivity is one of the pillars of socio-economic sustainability, and the manufacturing sector is of central importance for economic and social growth. The 8th objective of the 2030 Agenda for Sustainable Development Goal (SDG), adopted by all United Nations Member States in 2015, concentrates on achieving a higher economic productivity level. This can be accomplished by focusing on high value-added and labor-intensive sectors and attaining productive employment with equal pay for the equivalent value of work, which shall result in a sustainable labor market.

Globally, labor productivity was 1.9% in 2018, compared to 2.0% in 2017, reflecting a downward trend in the world's growth in output per worker from the 2.9% per annum average for 2000–2007 to 2.3% for 2010–2017 [2]. Malaysia has fallen behind most advanced countries in its labor productivity level. In 2019, for example, Malaysia's labor productivity was approximately half that of the United States and Singapore, based on terms of purchasing power parity. On the other hand, Malaysia exceeded its regional peers, including Indonesia, Vietnam, Thailand, China, and India. However, these countries have higher labor productivity growth than Malaysia, which means that they are rapidly catching up in labor productivity [3] (see Figure 1).

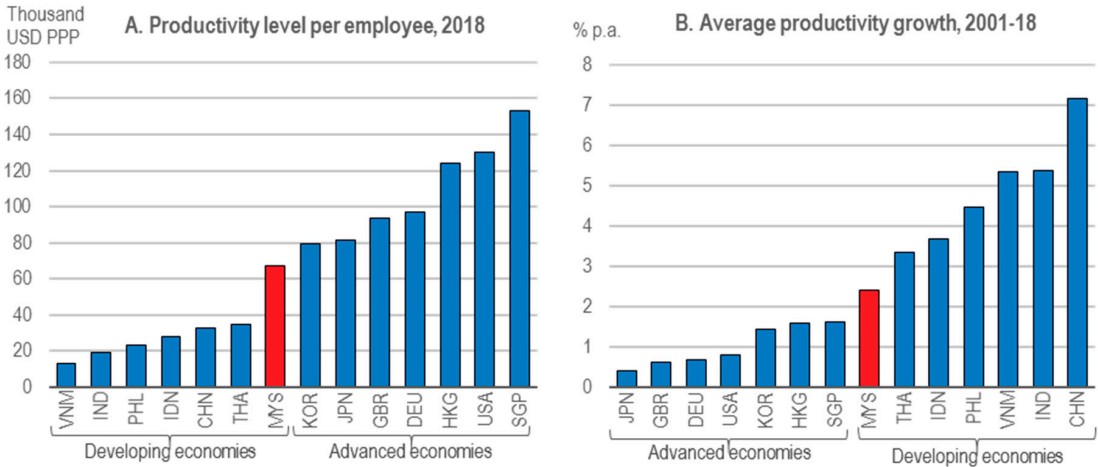

**Figure 1.** Labor productivity level still lags behind advanced countries. Source: OECD [3].

Malaysia has remained static in the middle-income trap for decades. Leaping from middle to high income is challenging. Higher labor productivity translates into a higher income per capita, hence a higher standard of living. In October 2018, the mid-term review of the 11th Malaysian Plan (11MP) announced that the target year to achieve high-income nationality status was delayed from 2020 to 2024 due to recent macroeconomic developments. With this new target to be delivered, Malaysia should focus on productivity growth with structural reforms to move up the value chain and improve labor skills [3].

Various empirical sources substantiated concerns over skill constraints. The development of human capital is one of the critical challenges for boosting work productivity in Malaysia. The National Industry 4.0 policy framework identified the future building workforce as the first national impetus and the upskilling of existing and future talented workers as one of the five strategic enablers. Malaysia currently depends heavily on low- and semi-skilled workers and foreign labor [4]. The situation arose ever since the Malaysian economy became focused on low value-added products over the years. However, the condition worsened as employers were reluctant to pay for skilled workers but relied instead on the pool of cheap low-skilled foreign and local workers. As a result, labor quality to economic growth remains much lower than the OECD average and the average of neighboring economics cluster, namely Indonesia and Thailand [3]. Many highly skilled Malaysians preferred working in other countries, which offered a better remuneration package. Skilled workers are vital to facilitate innovation and the adoption of technologies and promote the advancement of activities that unlock economic growth potential. Malaysia has launched an innovation-fostering program since the mid-1990s to transform the economy from an input-driven to a knowledge-oriented approach [5]. The labor productivity of Malaysia remains low, however, and the economy continues to be heavily dependent on factor inputs, particularly the accumulation of non-ICT resources. Consequently, Malaysia has remained behind most advanced countries in its labor productivity level.

While Malaysia is well endowed with natural resources, the manufacturing sector has played a crucial role in turning her into an essential player in the global value chain, apart from rapidly transforming it into an industrialized nation that continues to drive economic growth. Productivity increases may have transpired through increased worker effort in response to receiving better wages as postulated in the gift exchange model by Akerlof [6] or shirking model by Shapiro and Stiglitz [7]. However, Malaysia's premature deindustrialization process has undermined its manufacturing industry's capacity to experience a sustained increase in wages. In contrast, the successfully industrialized countries, namely, the United States, Germany, Japan, Korea, and Taiwan, succeeded in maintaining a positive real wage growth. Murugasu et al. [8] concluded that Malaysian workers receive lower wages than benchmark economies even after taking into account productivity differences. Furthermore, Malaysia has a smaller labor share of income despite its

labor-intensive nature, which is believed to be due to the slower pace of technological advancements and human capital development.

The Malaysian manufacturing industry, which relies heavily on foreign labor, faces continuous challenges to increase efficiency and productivity by adopting new technology. Imports of unskilled foreign labor and failure to stimulate technology upgrading weakened the manufacturing sector's capacity to support improvements in labor productivity and wage [9]. The increasing capital-labor intensity in manufacturing resulted in higher labor productivity through labor-replacing automation. However, the manufacturing sector has failed to strengthen, as it has not produced any world-class domestic technology firms [10]. Based on the 11th Malaysia Plan (2016–2020), the country embarked on a more challenging journey, targeting complex and diversified products that can contribute to high value-added performance [5,11]. The manufacturing industry has lower capital intensity and smaller labor income share than the benchmark and advanced economies of the US, United Kingdom, Australia, Germany, and Singapore [8]. Labor productivity is often influenced by the labor market situation and the size of investments in the economy. These investments increase the capital-labor ratio and, consequently, economic growth. Labor productivity shifts between sectors and industries and reflect recent events and economic conditions [12].

Despite all these challenges, many researchers have advocated the revival of the manufacturing sector as Malaysia's primary sources of jobs. Therefore, corresponding wages, appropriate capital intensity, and sufficient skilled labor are believed to boost labor productivity in the manufacturing industry. Basri et al. [13] found a positive relationship between wage, skilled labor, and capital intensity with labor productivity in the Malaysian manufacturing industry. However, there is limited information on how and to what extent the factors of production, namely, wage, capital intensity, and human capital, affect economic performance, given that all are essential drivers of labor productivity. A sector aggregate productivity is comprised of components of productivity for industries and the allocation of factors of production among these industries. Since these two components may interact, the shock in any production factors may influence productivity in other industries within the same sector.

Motivated by the above lacuna, this study attempts to bridge the gaps in present literature theoretically and practically. The novelty contribution of the study is in its estimate, an industry-dynamic way of the effects of wage shocks, capital intensity, and human capital on labor productivity. Some past studies have similarly focused on such relationships. The findings of this study should, to some extent, complement the past literature, which mainly discussed the direction of relationship or causality. To our best understanding, information on the impact upon labor productivity of shocks arising from the variables of interest in this study is almost non-existent. The effects of these factors on labor productivity in Malaysian manufacturing have not been sufficiently quantified, even though they are essential and well recorded in manufacturing and other industries' literature.

Furthermore, due to the increased economic importance of the above elements of production in Malaysia, the appropriate measurement of its impact is crucial for effective policymaking. Therefore, this study applies PVAR estimation, which allows for the analysis of the IRF for one variable shock on another. The role of impulse-response functions (IRF) is to capture the dynamic interactions among the endogenous variables. This would enable the study to forecast how a shock in any of the three selected production inputs will affect labor productivity. The IRF can account for any delayed effects on and of the variables under consideration and determine whether the effects between the variables of interest are short-lived, long-lived, or both. The typical panel regression estimators would not have captured such dynamic effects. The study also assesses the variance decomposition effect (FEVD) that explains the importance of one variable over another in the system within a given period, based on a PVAR. FEVD plays a role in elucidating the explanatory power of factors of production on labor productivity. Solving the missing puzzle in the existing body of knowledge is crucial to facilitate accurate policy formulation and solve misspecification issues.

Thus, this study aims to empirically examine the effects of factors of production shocks, as per the variables stated above, upon labor productivity of the manufacturing industry in Malaysia as

this should present relevant information to assist the policymakers. Besides this, the study further explains the extent of these impacts on the productivity of labor. Indeed, it has been a challenge for decades to identify the applicable policy to boost economic growth to free the nation's economy from the middle-income trap.

The remainder of the paper is organized as follows. Section 2 reviews the literature to establish the theoretical links between wage, capital intensity, human capital, and labor productivity. The next section introduces data sources used to examine the impact of production shocks on labor productivity and the methodology employed for the study. The findings are discussed in Section 4, while Section 5 concludes the study, following a short discussion of the key results, and evaluates consequent policy implications.

## 2. Review of Literature

This section consists of three subtopics that represent each of the production factors and how they relate to labor productivity.

### 2.1. Labor Productivity and Wage

Economists have extensively researched the wage-labor nexus in the past three decades, but the findings are mixed. The main results have variously shown the nexus to be either positive, negative, or non-monotonic. Some researchers have studied the wage-productivity nexus in both the developed and developing economies. Among the pioneers, Hall [14], Alexander [15], and Strauss and Wohar [16], for example, found a positive long-run relationship between real wage and productivity. Kumar et al. [17] suggested that real wage showed Granger-caused productivity in the long run, where a 1% increase in manufacturing sector real wage led to a rise in manufacturing sector productivity of between 0.5% and 0.8%. Baffoe-Bonnie and Gyapong [18] found that an increase in wages did encourage manufacturing workers to increase their productivity in the short-run. The causal effect from real wage to productivity is unidirectional in Turkey [19], while Dritsaki [20] found a causal relationship between the real wage and labor productivity in Romania.

A recent study by Karaalp-Orhan [21] found the importance of real wages in long-run labor productivity in the Turkish manufacturing industry. The empirical analysis in India confirmed the existence of a long-term relationship between wage and productivity and found the efficiency wage theory to be more appropriate than the marginal productivity theory. Another study by German-Soto and Brock [22] found a statistically significant and negative relationship between labor productivity and salary in the early period, but this eventually turned positive. Ozturk et al. [23] showed that wages had a positive effect on labor productivity, which implies that an increase in salary stimulates the workforce's belief that they are substantially paid for their work. This belief ultimately increases their trust and loyalty to the employer.

Alexander [15], Hall [14], and Wakeford [24], on the contrary, revealed that real wage exerted a negative impact on employee productivity in the short-run but not the reverse. Samargandi [25] examined the role of compensation, human capital, and capital stock in labor productivity in the context of Middle East and North Africa (MENA) countries and found that compensation was negatively associated with labor productivity. Surprisingly, however, in a study in Greece, Hondroyiannis and Papapetrou [26] found a rather vague impact on productivity. Millea [27] showed that efficiency wages were paid in Canada, Italy, and the United Kingdom, but no such wage-setting exists in Sweden, the United States, and France. However, in South Africa, Tsoku and Matarise [28] found that real wages did not Granger cause labor productivity at the macroeconomic level. A recent study by Reza [29] also achieved similar results for the manufacturing industries in Iran, where these two variables showed mutual adverse effects. Gneezy and Rustichini [30] established that the impact of wage (monetary compensation) and labor productivity (performance) was non-monotonic, indicating that the relationship is not linear.

In the Malaysian context, the literature on wage-productivity linkage is still limited. An earlier study by Lee-Peng and Yap [31] investigated wage formation in the Malaysian manufacturing industry

from 1975 to 1997. They found a significant positive relationship between wage and productivity for the manufacturing industry, where the increase in real wage exceeded the increase in labor productivity in the long run. Goh [32] and Goh and Wong [33] later found that real wages did not affect productivity. Conversely, Tang [34,35], using annual data from the manufacturing sector, and employing a Granger causality test, discovered that real wage Granger affects labor productivity. He found no evidence of reverse causation. Fahmy-Abdullah et al. [36] concluded that transport manufacturing firms in Malaysia should increase wage rates to improve employees' productivity and motivation. A recent study by Basri et al. [13] based on Pooled Mean Group (PMG), a dynamic panel data estimator, established that real wage exerted a significant positive impact on labor productivity in the short and long-run. The results confirmed the efficiency wage theory. The study further showed that an increase of 1 percent in real wage would increase labor productivity by 0.77% in the short-run but only 0.46% in the long-run.

### 2.2. Labor Productivity and Human Capital (Skilled Labor)

The number of skilled labor is often used as a measure of human capital. Skilled workers are a workforce segment with a high level of skill capable of creating significant economic value through the work conducted [37]. Such skill is generally characterized by elevated experience and expertise level befitting complicated tasks requiring specific skill sets, education, training, experience, and abstract thinking. The skilled worker requires some form of professionalism and training, which does not necessarily require a college degree or the equivalent [38]. Thus, an appropriate level of skill and training among the workers may increase their bargaining strength for more salaries, while encouraging them to produce more [39].

The literature extensively documented the role of human capital on performance. According to the neoclassical growth theory introduced by Lucas [40], economic progress is a function of the level of human capital development. Human capital here is defined from the perspectives of educational attainment and learning by doing. The learning process necessarily involves risks, uncertainties, and costs [41], where the "S" learning curve clearly explains the capabilities to acquire knowledge. The learning process is a function of the technology used. Some technologies are embodied, while others depend extensively on implicit knowledge. Knowledge is thus perceived as critical on performance.

In organizational studies, the resource-based view of firms links human capital to performance. Many scholars regard human capital as a core competency factor, according to this theory. Hamel and Prahalad [42] defined it as a valuable set of firms' assets and internal capabilities. This view is an extension of that initially conceived by Penrose [43] and later propounded by other workers based on their resource-based interpretations [44–48]. Collectively, they contended that firm-specific factors, including human capital, are essential. Firms with intensive human capital investments were empirically proven to enjoy superior profits [49]. Their studies also showed that investments in human capital strengthen firms' intangible assets, which is precisely the intellectual capital. The more intellectual capital invested, the higher the firm's competitive advantage to boost its products or processes above those of other competitors, thus allowing the firm to sustain its profits in the long-run.

According to Becker and Gerhart [50], many studies have shown that natural resources, technology, and achieving economies of scale are replicable factors. Still, human capital and its strategies are distinctive and flexible and, therefore, perfect. A human capital advantage could thus preserve a firm's competitive edge and increase its survivability. Skills and knowledge imbued in human capital are indeed the most critical factors for performance. Since human capital can be regarded as a vital driver, it follows that its effective capitalization should, in turn, increase the competitive advantage of firms [51].

In the Malaysian experience, past empirical studies similarly discovered that human capital plays a significant and substantive relationship with the firm's performance, irrespective of the type of industry [52]. The overall productivity in the National Key Economic Areas (NKEAs) was significantly boosted through a contribution from high-skilled labor [53]. In their study, which spanned 2000–2010,

Chandran et al. [11] revealed the vital role of skilled labor in increasing the value added across subsectors of manufacturing in local and foreign firms. The manufacturing sector should be transformed from labor- to capital-intensive, thus reducing low skilled work and enhancing the manufacturing sector's contribution to the Gross Domestic Product (GDP) [54]. Human capital was also proven to significantly increase Malaysian SMEs' labor productivity, as gauged through training expenses as a proxy [55].

### 2.3. Labor Productivity and Capital Intensity

The study by Jain [39] suggested that capital intensity should match the skill intensity. The capital intensity was positively associated with labor productivity in a study conducted for MENA countries [25]. Capital-intensive firms, by nature, necessitate the advantages of economies of scale.

Higher productivity becomes a crucial factor for survival since new industry players are forced to make major investments in facilities, equipment, and other assets [56]. Chauvin and Hirschey [57] argued that larger firms tend to have lower transaction costs. Large firm size translates into greater production capacity and higher productivity. Firms in capital-intensive industries, such as chemicals or aluminum production, shipbuilding or aeronautics, have excess capacity that may potentially lead to damaging price competition. Often it becomes too difficult and expensive for such firms to manage the closure of any facility [58]. Such argument concurs with the fundamental principles of the resource-based view. Firms with high capital intensity tend to concentrate on exploiting their investments [59], thus indirectly leading to a greater concern for costs and efficiency in realizing an increase in labor productivity.

Erenburg [60] studied the long-run linkage between labor productivity and real wage in the United States. He discovered a counter-cyclical relationship between the real wage and labor productivity once the empirical stance had control over capital stocks. Real wage will raise the unit cost of labor at the macro-economic level, thus inducing substitution from labor to capital and consequently increasing marginal (and hence average) labor productivity [24].

According to Jajri and Ismail [61], capital stock and capital-labor ratio contribute significantly to Malaysian economic growth and labor productivity, respectively, whereas productive labor contributes less than physical labor. Turner et al. [62] concluded that a more extensive stock of physical capital per worker creates a more productive economy and also leads to increased labor productivity. The net fixed assets or net capital stock divided by employment represents the capital-labor ratio, contributing to an accelerated labor productivity growth rate. A study in 2015 using data from 531 Malaysian firms in the electrical and electronic manufacturing industry established that capital-labor ratios show a positive relationship with technical inefficiencies in firms [63]. While efficiency is linked to productivity, long-term efficiency can help increase productivity growth [64].

## 3. Data and Methodology

### 3.1. Data Description

Data used in this research are derived from the Annual Survey of Manufacturing Industries by the Department of Statistics Malaysia (DOSM) for the period 2000 to 2015, covering 44 industries. All variables employed in this study are the unpublished data, which means that they are only available upon request to DOSM, thus not publicly available on any website. There was a change in the industrial classification from the Malaysia Standard Industrial Classification (MSIC) (2000) to MSIC (2008). As such, this required the data to be aggregated into the 3-digit MSIC level for the ensuing empirical inquiry. This study adopted the industry classification concordance table provided by the DOSM to match the industries.

Log-difference was employed for the variables of interest. Capital intensity is represented by the log-difference of the real capital intensity value ($\Delta$lkl), calculated as the capital per total employees (in RM'000). Human capital denotes the percentage of skilled workers (professionals, executives, technicians, and supervisors) per total employees ($\Delta$npt). This study used the percentage of skilled

labor per total employees as the measure for human capital, in keeping with past literature. Wage, indicated by log-difference of the real wage value (Δlwl), represents the real wage per total employees (in RM′000), and labor productivity, which is the log-difference of the real labor productivity (Δlyl), refers to the real value-added per total employees (in RM′000).

### 3.2. Methodology

There are five econometrics techniques for this study: (i) cross-sectional dependence, (ii) panel unit root test, (iii) panel causality test, (iv) PVAR analysis comprising IRF and FEVD, and (v) robustness analysis. This study conducts a rigorous robustness analysis using a proxy for the percentage of skilled labor, different lag, and various instruments.

### 3.2.1. Cross-Sectional Dependence and Unit Root Test

Pre-analysis checks on panel data are crucial before choosing the most appropriate estimation method. In this context, the preliminary step is to determine cross-sectional dependence (CD) in the selected panel data and the order of integration before estimating the IRF and FEVD. On the possibility of CD occurring in panel data, it is categorical that employing conventional panel unit root tests may result in incorrect test results [65]. Thus, this study applies the Pesaran [66] CD test before testing panel data's stationarity. The data's stationarity is then assessed by the second-generation cross-sectional augmented (CIPS) panel unit root test of Pesaran [66] after cross-sectional dependence in the panel data is confirmed.

### 3.2.2. Panel Causality Test

The newly introduced Dumitrescu-Hurlin (DH) [67] non-causality test is a suitable method for this study as it offers more advantages than the traditional Granger [68] causal test. The DH test considers the heterogeneity of the regression model and the heterogeneity of the causal relationship, two dimensions of heterogeneity, in addition to taking fixed coefficients like the Granger causality test.

### 3.2.3. Panel VAR: Impulse Response and Variance Decomposition Analysis

This study employs a PVAR methodology developed by Love and Zicchino [69] in a generalized method of moments (GMM) framework to account for dynamic endogeneity. PVAR technique is favored as it has the features of the traditional VAR model that considers all variables as endogenous in the panel-data estimation approach. The latter indicates that each cross-sectional dimension has fixed effects that allow controlling for individual firm-level heterogeneity. The residuals in the equations are disturbances that are uncorrelated over-time and are then exploited through (i) identification of a recursive structure (in a matrix), and (ii) estimating the response of the variables to structural shocks (impulses). Following Andrews and Lu [70], the optimal lag for the model selection was based on the first-order PVAR. This model can be expressed as:

$$Y_{it} = \mu_i + \Phi(I)Y_{it-1} + f_i + \varepsilon_{it} \tag{1}$$

where, $Y_{it}$ is a vector of endogenous variables, $\Phi(I)$ is the polynomial in the lag operator, and $f_i$ is a vector of industry-specific effects. $\varepsilon$ signifies $Y_{it}$ which comprises labor productivity (log-differences) of the following three endogenous variables: capital intensity, which refers to real capital per worker (Δlkl); human capital, which refers to the percentage of skilled labor (Δnpt); and real wage per labor (Δlwl). Lastly, $\varepsilon_{it}$ denotes a serially uncorrelated error term with zero means, and also known as structural shocks.

The study will use the orthogonal impulse-response shocks to analyze the effects of the industry-specific factors on labor productivity. It is necessary to identify short-run model restrictions using the conventional Cholesky decomposition technique in VAR models in applying the recursive structure for orthogonalization of impulse-responses [71]. The Cholesky decomposition of the

variance-covariance matrix is used to identify orthogonal shocks in the interest variables and examine their effect on the system's remaining variables while holding other shocks constant. To analyze one variable's response to an orthogonal shock in another variable, we focus on impulse-response functions (IRFs). The confidence intervals for the orthogonalized IRFs are generated with Monte Carlo simulation. The shocks are identified by assuming a recursive structure. Variables that enter first in Equation (1) are assumed to be the most exogenous and hence affect subsequent variables both contemporaneously and with a lag. In contrast, variables that are ordered later are less exogenous and will only affect previous variables only with a lag. This study assumes that capital intensity is the most exogenous, where the nature of business has been pre-determined upon establishment. In contrast, human capital and wage are less exogenous as human capital is very much dependent on the type of industry. In contrast, wage is usually determined by the type of work that is based on the skill needed by the industry. The productivity, namely, the real value-added per employee (*lyl*), is hence set on.

The following matrix represents the Cholesky decomposition for this study:

$$
\begin{pmatrix} \varepsilon_{it}^{lkl} \\ \varepsilon_{it}^{npt} \\ \varepsilon_{it}^{lwl} \\ \varepsilon_{it}^{lyl} \end{pmatrix} = \begin{bmatrix} a & 0 & 0 & 0 \\ b & c & 0 & 0 \\ d & e & f & 0 \\ g & h & i & j \end{bmatrix} \begin{pmatrix} e_{it}^{1} \\ e_{it}^{2} \\ e_{it}^{3} \\ e_{it}^{4} \end{pmatrix} \tag{2}
$$

According to Cholesky, numbers above the diagonal are set equal to zero, and those below are the free parameters. In the above model, *a*, *c*, *f*, and *j* diagonals represent their shocks in the system, while *b*, *d*, *e*, *g*, *h*, and *i* are the free parameters. In Equation (2), The error terms or structural shocks are isolated variables in the system (recursive structure), which become orthogonal as explained by the alphabetically ordered weights connected to the model's structural shocks while maintaining all other responses at zero value. In this concept, it is assumed that the variables at the front of the ordering pattern will influence other factors contemporaneously and with an exogenous lag, whereas delayed factors will affect the front variables with an endogenous lag.

The assumption specified for Model (2) is that the instantaneous shocks in lkl, npt, and lwl have a contemporaneous effect on lyl (real labor productivity). Besides this, lyl only exerts a lagged effect on the lwl, npt, and lkl. All variables are selected based on the Cobb–Douglas production function to denote production in manufacturing industries with the technology, capital, and differentiated labor as factors of production [13,72,73].

Each industry-level heterogeneity was controlled by introducing fixed effects. However, due to the lags in the dependent variable, the fixed effects are correlated with the regressors, thus producing a bias in our PVAR coefficients. The bias can be overcome by using forward mean differencing or a Helmert transformation [74] as an alternative to the norm of using standard differencing from past observations. This technique subtracts the means of all available future observations in each firm-year. In this way, the orthogonality between transformed variables and lagged regressors was preserved. Further, it allows for identification of the model in which lagged regressors are used as instruments, and the coefficients of the PVAR to be estimated by the system GMM.

A shock is an economic term designating an unexpected or unpredictable event that affects an economy, either positively or negatively. The event was measured using the IRF, which described the reaction of one variable to the shock emanating from another variable within a system, while all shocks were simultaneously held at zero value [69]. The statistical significance of the IRF was gauged through standard errors of the estimated PVAR coefficients taken with their variance-covariance matrix. The confidence intervals were then generated by using the 200 Monte-Carlo simulations. The FEVD that estimate the percentage variation in each variable caused by the shock to another variable (accumulated over time) was recorded over a time window of 10 periods (years).

## 4. Results and Discussion

### 4.1. Cross-Sectional Dependence and Panel Unit Root

The previous studies in labor economics regarding production literature have not addressed cross-sectional dependence in the analysis. Therefore, the second-generation cross-sectional augmented (CIPS) panel unit root test of Pesaran [75] is used to determine the degree of integration of our series of interest in our panel of 44 manufacturing industries.

Table 1 shows that the null hypothesis of cross-sectional independence is rejected, as the *p*-values are close to zero. The Pesaran [66] CD test strongly rejects the null hypothesis of no cross-sectional independence for all variables (lkl, npt, lwl, lyl).

**Table 1.** Cross-sectional dependence test.

| Variables | CD Test | *p*-Value |
|---|---|---|
| labor productivity (lyl) | 10.87 | 0.000 |
| capital per labor (lkl) | 25.30 | 0.000 |
| wage per labor (lwl) | 29.24 | 0.000 |
| % of skilled labor (npt) | 13.86 | 0.000 |

Having established that all the series are cross-sectional correlated, the next step is to implement a panel unit root test that accounts for the presence of cross-section dependence. One of the tests is the cross-sectionally augmented version of the IPS test, the CIPS (or CADF) test. CIPS unit root test results as shown in Table 2 confirmed the presence of unit root, that is, the data are stationary at order one [I(1)].

**Table 2.** Panel unit root test results.

| Variables | CIPS (CIPS Statistic) |
|---|---|
| labor productivity (level) | −2.402 |
| labor productivity (first difference) | −3.442 |
| capital per labor (level) | −1.965 |
| capital per labor (first difference) | −3.183 |
| wage per labor (level) | −2.582 |
| wage per labor (first difference) | −4.119 |
| % of skilled labor (level) | −2.504 |
| % of skilled labor (first difference) | −3.723 |

Notes: The Stata routines xtcips written by Sangiácomo [76] were used to estimate the CIPS statistics by Pesaran [75]. Deterministic chosen: constant and trend. Critical values for the CIPS statistics at the 10%, 5%, and 1% significance level are −2.63, −2.71, and 2.85 respectively for variables at the level, and 2.66, 2.76, and 2.93, respectively for variables at first difference.

### 4.2. Dumitrescu-Hurlin (DH) Causality

Table 3 reports DH panel causality results. The findings show that significant to reject all null hypotheses indicates that all the production variables, namely, capital intensity, human capital, and wage, are significant to Granger-cause labor productivity. These findings are in line with some previous studies in the Malaysian context, for example, Basri et al. [13], Chandran et al. [11], and Jajri and Ismail [61].

**Table 3.** Dumitrescu-Hurlin (DH) panel causality test.

| Null Hypothesis | Zbar-Statistic | *p*-Value |
|---|---|---|
| capital per labor *does not cause* labor productivity | 6.1359 *** | 0.0000 |
| % of skilled labor *does not cause* labor productivity | 2.5796 *** | 0.0099 |
| wage per labor *does not cause* labor productivity | 6.4543 *** | 0.0000 |

Notes: The Stata routines xtgcause written by Lopez and Weber [77] were used to estimate the DH statistics by Dumitrescu and Hurlin [67]. '***' denotes rejection of the null hypothesis at 1%. Lag 1 is chosen based on criteria established by Andrews and Lu [70] and Abrigo and Love [78] as per Table 4.

**Table 4.** Lag-order selection statistics for panel vector autoregression (PVAR) estimated using generalized method of moments (GMM).

| Lag | CD | J | J *p*-Value | MBIC | MAIC | MQIC |
|---|---|---|---|---|---|---|
| 1 | 0.83929 | 39.14617 | 0.81568 | −253.019 | −56.85383 | −134.2412 |
| 2 | 0.85574 | 23.14587 | 0.87369 | −171.630 | −40.85413 | −92.44571 |
| 3 | 0.50961 | 14.01124 | 0.59768 | −83.3771 | −17.98876 | −43.78455 |

### 4.3. Impulse Response

The estimated PVAR parameters do not provide much information, according to Galariotis et al. [79]. The moving average (MA) representation of the VAR model, namely, the IRF and the FEVD, should be focused. Therefore, these IRF and FEVD provide valuable information in achieving the focal point of this study, which examines how labor productivity responds to a surprise shock for all variables in the system. Following some previous research [79,80] that did not discuss the estimated PVAR parameters, we thus only report the IRF and FEVD. However, the full result of the PVAR coefficient is available upon request.

The next step is to determine the selection of lag for the PVAR model. For this, we use the overall coefficient of determination (CD) and the Moment and Model Selection Criteria (MMSC) developed by Andrews and Lu [70]. The evidence shown in Table 4 is supportive of the choice of one lag. This study employed the first- to third-order PVAR using the first four lags of kl, npt, and lwl as instruments to calculate the selection structure. According to the established criteria by Andrews and Lu [70] and Abrigo and Love [78] about the MMSC and the overall CD, the first-order PVAR model is favored among the three lag orders based on the minimum values presented for the Bayesian information criterion (MBIC), the Akaike information criterion (MAIC), and the Hannan-Quinn information criterion (MQIC). Andrews and Lu's MMSC are based on Hansen's J statistic, which requires the number of moment conditions to be greater than the number of endogenous variables.

This study reports evidence of the stability properties of the estimated PVAR model in Table 5 and Figure 2. The stability of the PVAR requires the moduli of the dynamic matrix's eigenvalues to lie within the unit circle, and our estimated model satisfies this condition. Referring to the eigenvalues (below one) in the estimated model, the PVAR model was found to satisfy the stability condition that presents invertibility and infinite-order vector moving-averages (VMA). Both Brüggemann et al. [81] and Hamilton [82] implicated VAR stability in every instance wherein every modulus in the companion matrices returned values lower than one.

**Table 5.** Results of the eigenvalue stability condition.

| Eigenvalue | | |
|---|---|---|
| Real | Imaginary | Modulus |
| −0.343718 | 0 | 0.343718 |
| −0.234300 | 0 | 0.234300 |
| −0.152667 | 0 | 0.152667 |
| −0.006216 | | 0.006216 |

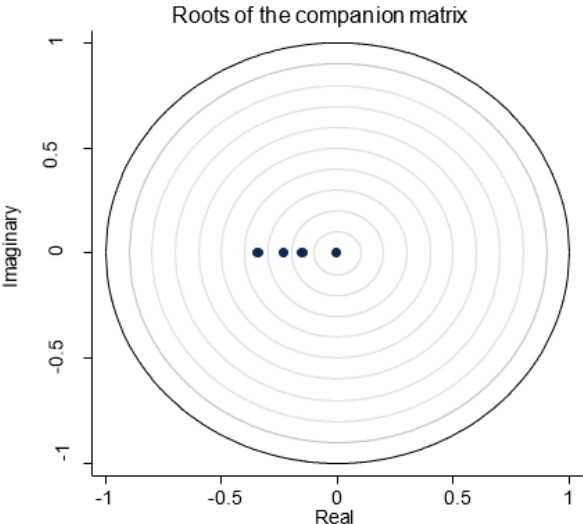

**Figure 2.** Roots of the companion matrix. Notes: The Panel VAR's stability requires the moduli of the eigenvalues of the dynamic matrix to lie within the circle unit. PVAR satisfies stability conditions as all eigenvalues lie inside the unit circle.

Subsequently, the impulse-response functions (IRF), which take into account contemporaneous as well as lagged responses [83] are presented in Figure 3 to capture dynamic interactions among the endogenous variables. The variables' ordering is as follows: capital intensity, human capital (skilled labor), wage, and labor productivity. The shocks' responses are significant when the plotted 95% confidence intervals, using 200 Monte Carlo replications, do not contain 0. Figure 3 illustrates the IRF when there is a shock in production factors on labor productivity, which shows an expected positive and significant response to a one standard deviation shock following the change of wage.

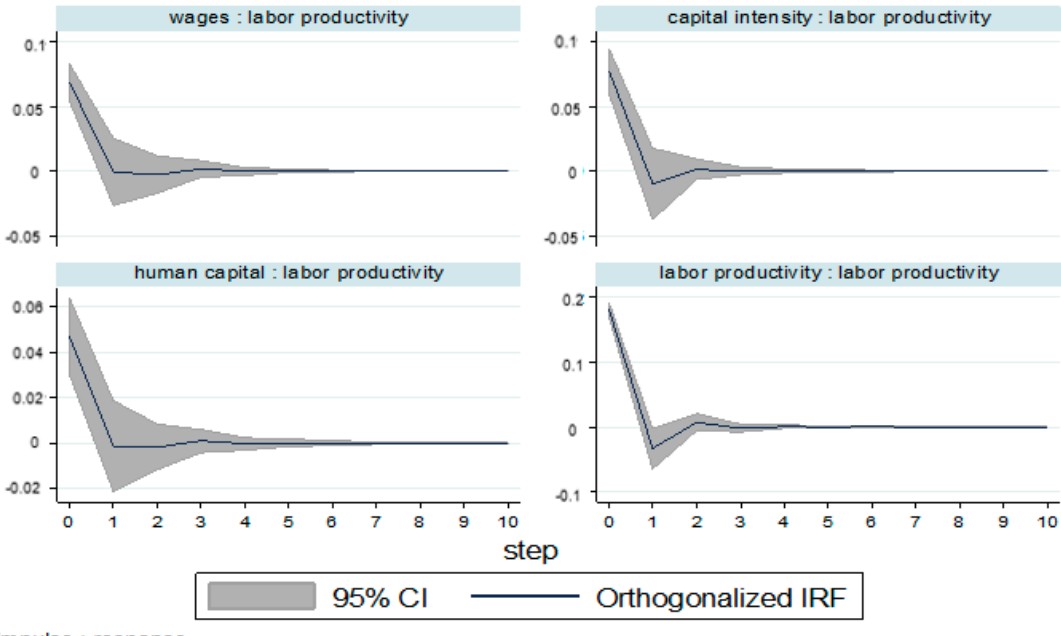

**Figure 3.** Orthogonalized impulse-response functions. Notes: The impulse-response function (IRF) was computed from an estimated PVAR (Equation (1)). One-standard error bands are based on 200 Monte Carlo simulations.

The above figure demonstrates that the positive shock will increase labor productivity growth contemporaneously by 0.07 basis point but will later marginally decay within one year and eventually stabilize in the subsequent years. The theory of deferred compensation [84] assumes that workers and firms want to be engaged in long-run relationships and concludes that rising earnings do not necessarily fully reflect increased productivity. Negative supply-side shock can also be one factor where an increase in wages will decrease output in the industry. Nonetheless, the economic impact of wage shock at 65.72% is substantial. The figure is obtained by comparing the one standard deviation response in shock to a standard deviation from Table 6.

**Table 6.** Results of descriptive statistics.

| Variable | Observations | Mean | Standard Deviation | Minimum Value | Maximum Value |
|----------|--------------|------|--------------------|---------------|---------------|
| Δlkl | 660 | −0.00569 | 0.34999 | −3.60582 | 3.26284 |
| Δnpt | 660 | −0.00584 | 3.74691 | −35.7587 | 28.0139 |
| Δlwl | 660 | 0.01477 | 0.10456 | −0.76893 | 0.63294 |
| Δlyl | 660 | 0.02216 | 0.34952 | −3.42834 | 2.51773 |

Note: The annual data is obtained from the Annual Survey of Manufacturing Industries published by the Department of Statistics Malaysia (DOSM).

While the positive shock of one standard deviation of change in capital intensity will increase labor productivity growth by 0.08 basis point contemporaneously, the shock associated with skilled workers change may only raise labor productivity by half of the amount, which is 0.05 basis point. The economic impact of capital intensity shock is rather moderate at 22.03%, and there is only a small impact on labor productivity from the shock in human capital. Similar in trend to the wage shock, the effect on labor productivity growth is positive, but only short-lived (less than 2 years) since it becomes insignificant thereafter. The findings on the magnitude of the economic impacts of the three factors towards labor productivity complement the results in the study by Basri et al. [13], which used the same set of data and found a positive relationship between the production factors and labor productivity.

Similar results apply when the aspect of levels is examined instead of differences. Figure 4 depicts the cumulative IRF from the PVAR. By cumulating the impact over time, these line plots show the effect on level form, rather than on the differences of variables and labor productivity (both in log forms). Although the line plots appear dissimilar in Figure 3, the interpretation is similar. A shock to wage (top-left), capital intensity (top-right), and human capital (bottom-left) commonly exert a significant positive impact on productivity. Nonetheless, the responses to the shocks have been stationary as they are close to zero.

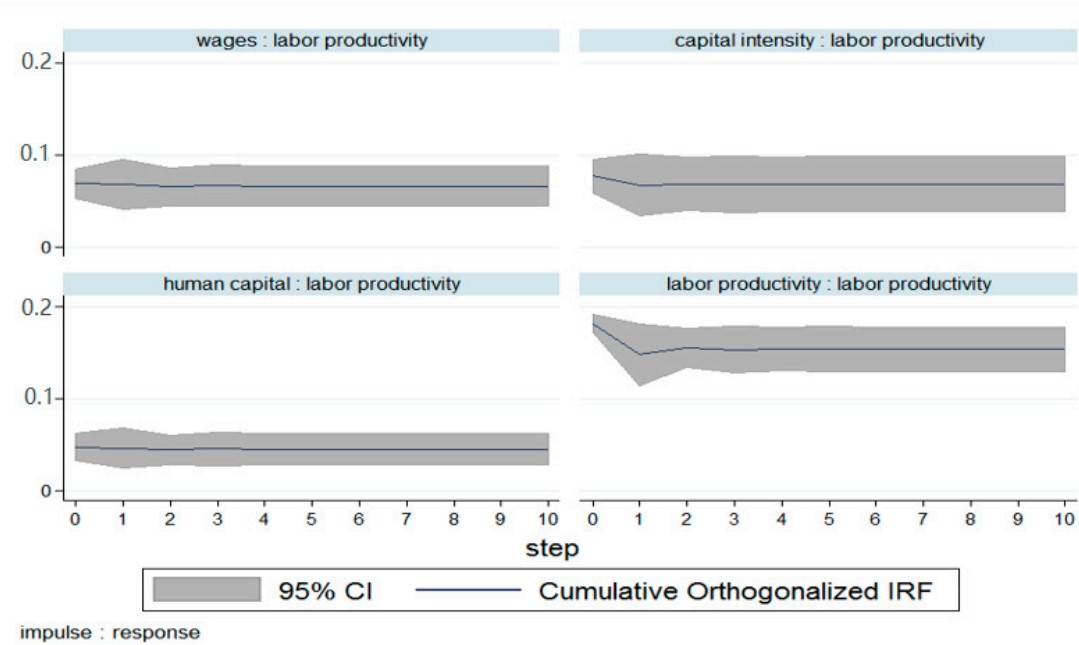

**Figure 4.** Cumulative orthogonalized impulse-response functions. Note: One-standard error bands are based on 200 Monte Carlo simulations.

## 4.4. Variance Decomposition

To assess the amount of information each production factor contributes to labor productivity over time, FEVD is calculated for the baseline PVAR model. FEVD shows how much of the future uncertainty of one time series is due to future shocks onto the other time series in the system. Since this evolves, the shocks on time series may not be very important in the short-run but very important in the long-run. As shown in Table 7, capital intensity (Δlkl) has the largest explanatory power for labor productivity (Δlyl), explaining about 13% of the total variance in labor productivity, followed by wage (Δlwl) with a close figure of 10%, whereas human capital (Δnpt) explained only 5%.

**Table 7.** Results of forecast error variance decompositions.

| Response Variable and Forecast Horizon | | Impulse Variable | | | |
|---|---|---|---|---|---|
| | | Δlkl | Δlwl | Δnpt | Δlyl |
| | 0 | 0 | 0 | 0 | 0 |
| | 1 | 0.130125 | 0.048691 | 0.10336 | 0.717823 |
| | 2 | 0.128882 | 0.047449 | 0.10065 | 0.723018 |
| | 3 | 0.128732 | 0.047445 | 0.100689 | 0.723134 |
| | 4 | 0.128715 | 0.047457 | 0.100715 | 0.723114 |
| Δlyl | 5 | 0.128713 | 0.047459 | 0.100719 | 0.72311 |
| | 6 | 0.128713 | 0.047459 | 0.100719 | 0.723109 |
| | 7 | 0.128713 | 0.047459 | 0.100719 | 0.723109 |
| | 8 | 0.128713 | 0.047459 | 0.100719 | 0.723109 |
| | 9 | 0.128713 | 0.047459 | 0.100719 | 0.723109 |
| | 10 | 0.128713 | 0.047459 | 0.100719 | 0.723109 |

Notes: The table reports the 10-years-ahead forecast error variance of labor productivity explained by a one s.d. The shock of the column variables. The Cholesky ordering used is Δlkl→Δnpt→Δlwl→Δlyl. The FEVD standard error and confidence intervals are based on 200 Monte Carlo simulations.

These calculations demonstrate that the weightage of capital over labor and wage has a fairly large explanatory power for labor productivity over the 10 years.

### 4.5. Robustness

The rigorous robustness analyses using (a) a proxy for one production factor, (b) other lag, and (c) alternative instrument are performed to ascertain whether the conclusions of this study are robust. The proxy used for the percentage of skilled labor is skilled labor over the non-skilled one. Lag 2 is tested for the robustness test, while lag 1 is employed for the PVAR baseline model. The labor productivity growth shows an expected positive and significant response to a one-standard-deviation shock in change for all three production factors but is slightly lower in magnitude in response to shock due to the growth of wage. When a different instrument is used, a parallel trend is obtained with a smaller contemporaneous effect on labor productivity following the shock of human capital change. Overall, these results are robust relative to those of the PVAR baseline model discussed earlier. See Figures 5–13.

(a)    using a proxy for the percentage of skilled labor

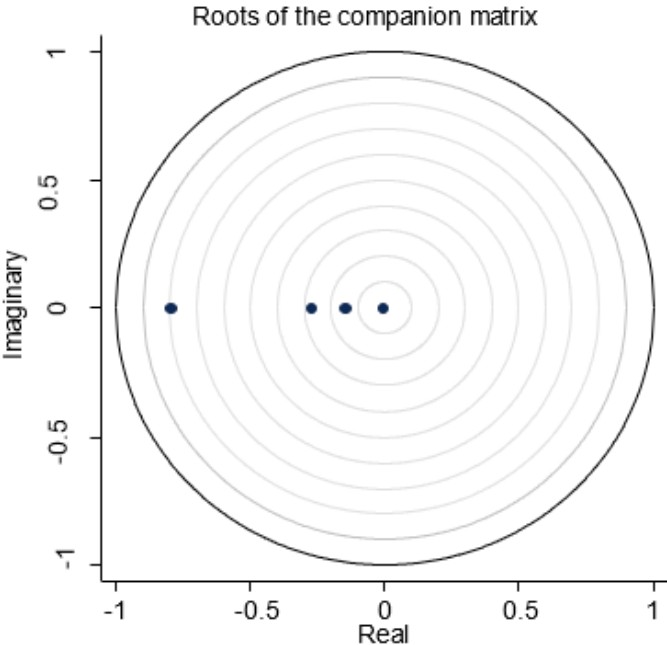

**Figure 5.** Roots of the companion matrix. Note: The Panel VAR's stability requires the moduli of the eigenvalues of the dynamic matrix to lie within the circle unit. PVAR satisfies stability conditions as all eigenvalues lie inside the unit circle.

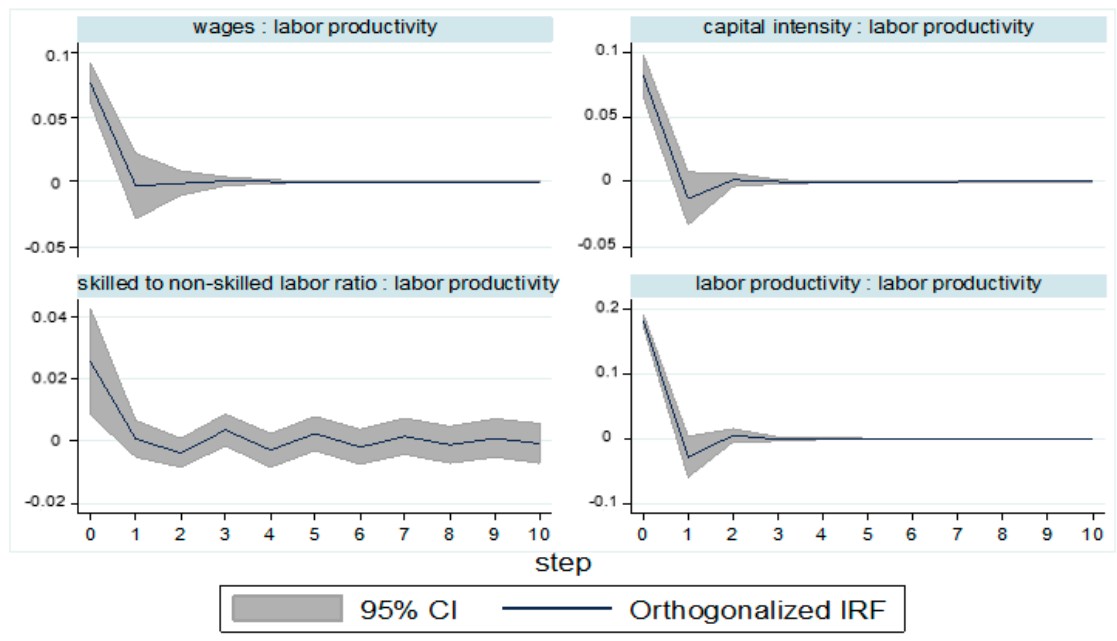

**Figure 6.** Orthogonalized impulse-response functions. Notes: The IRF was computed from an estimated PVAR (Equation (1)). One-standard error bands are based on 200 Monte Carlo simulations.

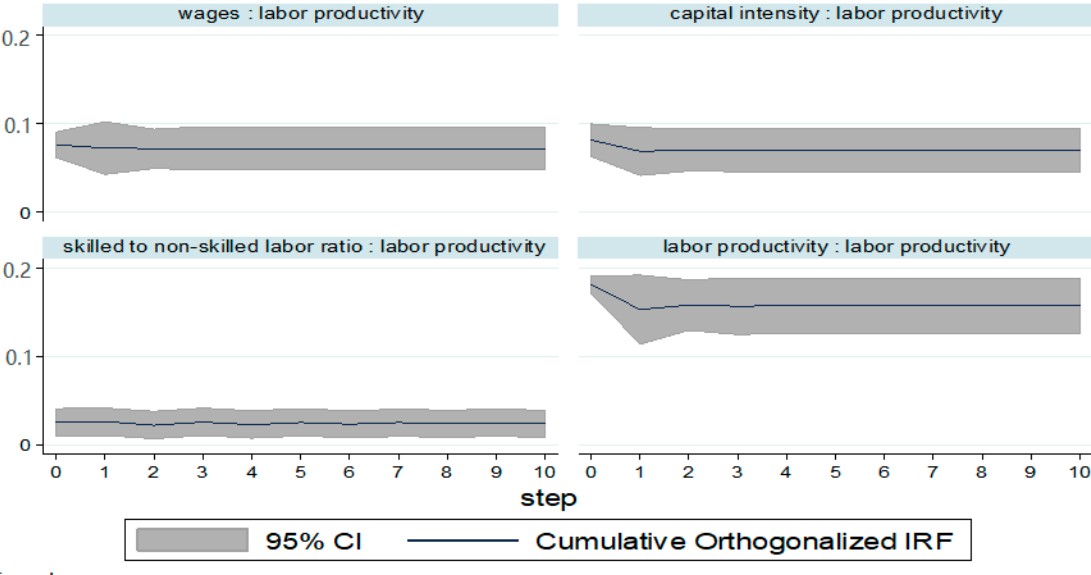

**Figure 7.** Cumulative orthogonalized impulse-response functions. Note: One-standard error bands are based on 200 Monte Carlo simulations.

(b)　using different lag (lag 2)

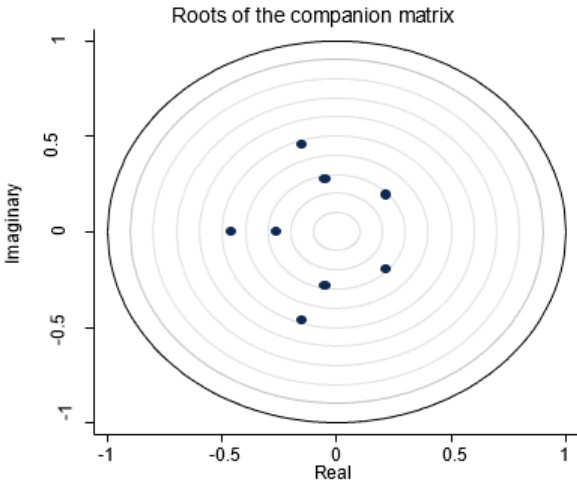

**Figure 8.** Roots of the companion matrix. Notes: The stability of the Panel VAR requires the moduli of the eigenvalues of the dynamic matrix to lie within the circle unit. PVAR satisfies stability conditions as all eigenvalues lie inside the unit circle.

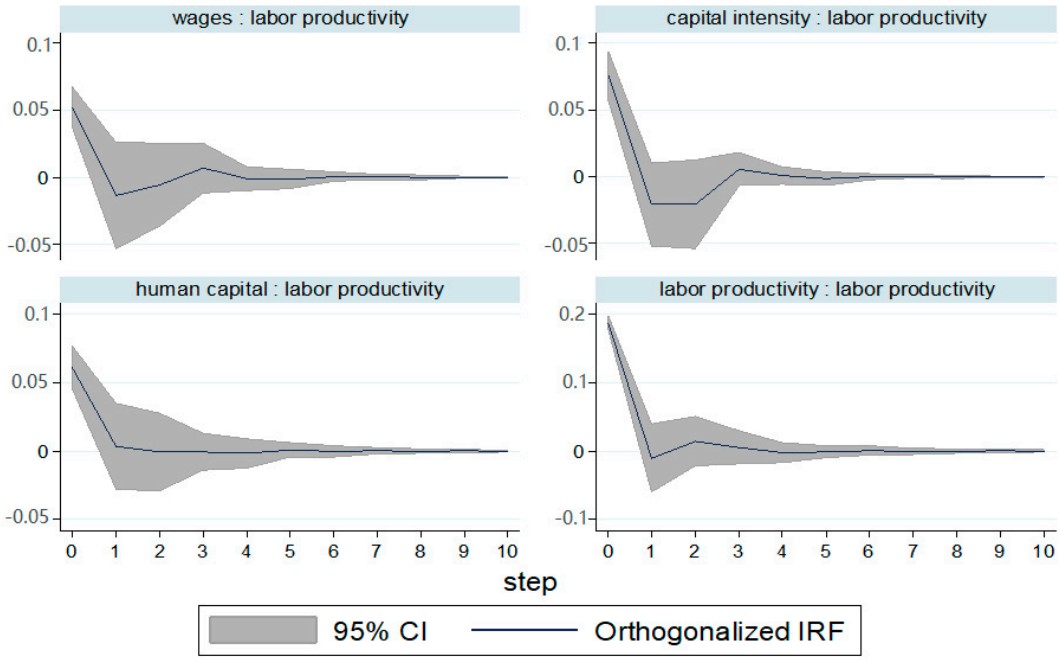

**Figure 9.** Orthogonalized impulse-response functions. Notes: The IRF was computed from an estimated PVAR (Equation (1)). One-standard error bands are based on 200 Monte Carlo simulations.

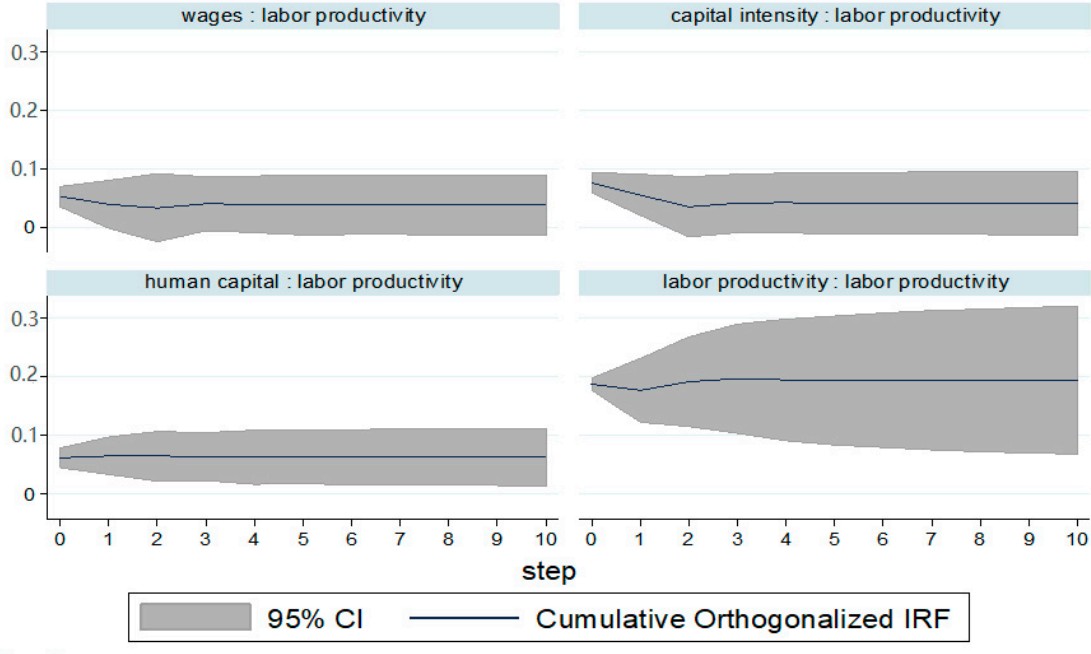

**Figure 10.** Cumulative orthogonalized impulse-response functions. Note: One-standard error bands are based on 200 Monte Carlo simulations.

(c)    using different instrument (instl(1/3))

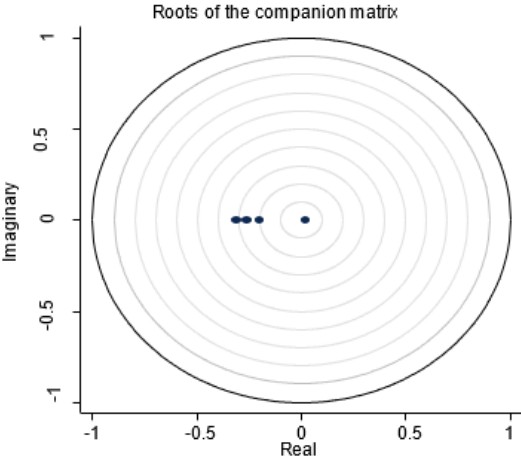

**Figure 11.** Roots of the companion matrix. Notes: The Panel VAR's stability requires the moduli of the eigenvalues of the dynamic matrix to lie within the circle unit. PVAR satisfies stability conditions as all eigenvalues lie inside the unit circle.

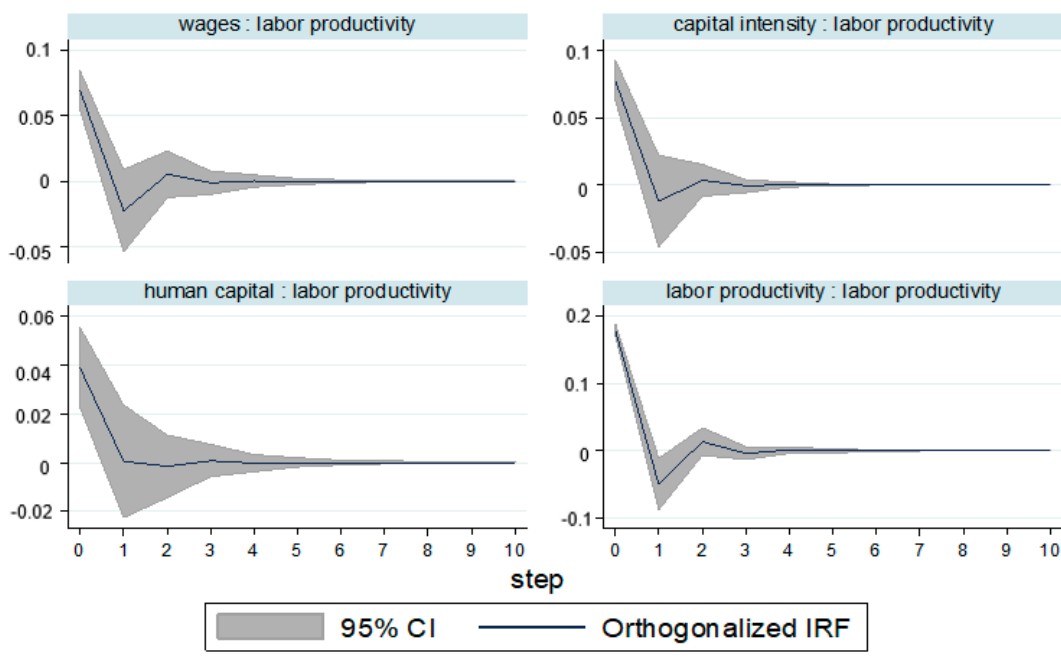

**Figure 12.** Orthogonalized impulse-response functions. Notes: The IRF was computed from an estimated PVAR (Equation (1)). One-standard error bands are based on 200 Monte Carlo simulations.

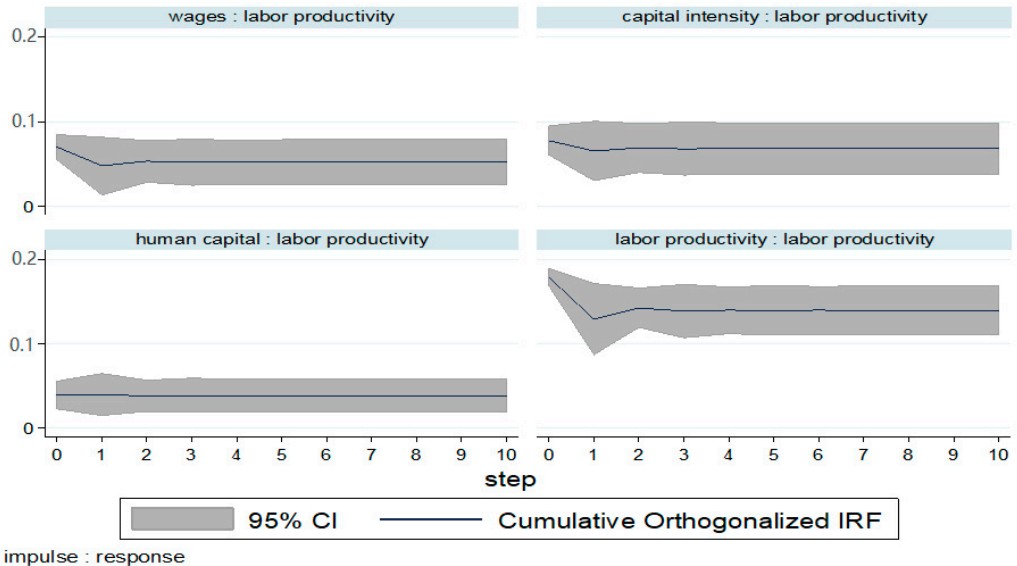

**Figure 13.** Cumulative orthogonalized impulse-response functions. Note: One-standard error bands are based on 200 Monte Carlo simulations.

## 5. Conclusions

The study examines how shocks in wage, capital intensity, and human capital affect labor productivity in Malaysia's manufacturing sector. For this purpose, the panel vector autoregression (PVAR) model was adopted as the methodology. The main finding is labor productivity growth, which shows an expected positive and significant response to one standard deviation shock in the change of wage, capital intensity, and human capital (refer to the skilled labor). The economic impact of a shock in wage growth on labor productivity is substantial at 65.72%, and the effect of shock in a change of capital intensity on labor productivity is moderate at 22.03%. However, labor productivity is minimally impacted by the shock in skilled labor change, at only 1.26%.



Apart from the impulse response analysis results, the amount of information each production factor contributes to labor productivity in the future varies. Capital intensity and wage have the largest explanatory power for labor productivity, clarifying about 13% and 10% of the total variance in labor productivity, respectively. Human capital recorded only 5% of explanatory power. These forecast error variance decomposition (FEVD) proves that the weighting of capital against labor and wage has a fairly large explaining power over labor productivity over ten years.

These novel empirical results augment the existing body of knowledge in the labor market theoretically by supplementing the findings in a study by Basri et al. [13] on the relationship between the production factors and labor productivity. The Malaysian Productivity Corporation has indicated in their Productivity Report for 2018/2019 that investment in capital must complement skilled labor and better remuneration to measure improvements in the qualitative aspects of labor and capital inputs are crucial in achieving higher productivity.

This study proposes four recommendations supporting the aspiration for the 8th objective in the Sustainability Development Goals (SDG), which is to achieve labor productivity sustainability, particularly for Malaysian manufacturing industries. *First*, it is to formulate policies on incentives and rewards for workers to motivate them to achieve their maximum work capacity. Given that the most significant impact of wage shock is on labor productivity, increases in wages are supportive of the Efficiency Wage Theory, which will elevate labor productivity. Employers are expected to pay high wages, which should motivate higher productivity of labor. The government should also review compensation occasionally to match the response to the productive sector's needs and improve the well-being of the labor force. The way forward is to formulate policies that may commensurate wages with the productivity level by enhancing the Productivity-Linked Wage System (PLWS) and reward system in human resource management.

Second, improve labor policies that relate to labor efficiency and productivity. These can be improved by examining per unit costs among inputs, increasing physical capital per worker and human capital per worker, and adopting new technology in eliminating weaknesses in labor law, wages system, management, and the reward system. Firms will improve the inputs' efficiency using per-unit costs by making appropriate adjustments of all inputs (labor, capital, and material inputs).

Third, by adopting new technology, firms can optimize production. Elevated automation levels promise higher production speed and precision that can assist productivity from stagnating in recent decades and provide room for more value-added inputs. The government is anticipated to give the manufacturers lucrative stimuli to realize such responses. Innovative methods are crucial in boosting labor productivity through research and development to amplify technology to improve efficiency.

Fourth and last, encourage industrial firms to hire more skilled labor to improve on the quality of human capital crucial in hastening productivity. Highly automated machines require skilled operators both in management and at the production lines to minimize loss due to human errors, such as in poor planning and imprecise execution of production procedures.

These efforts need to be efficiently integrated with the revolutionary concept of Industry 4.0 with the promise of stimulating productivity. However, this can only be achieved with the appropriate technologies and consummate skill sets to be complemented with attractive remuneration to motivate the workforce to stay focused and prolific in the industry. Other key contributors, for example the service and construction sectors, can be included in future research to understand better the impact of shocks on labor productivity in the Malaysian context. The response of labor productivity on other production factors shall also be taken into account for a more comprehensive understanding.

**Author Contributions:** N.M.B. carried out the experiment, wrote, and revised the manuscript with support from Z.A.K. and N.S. The central idea of this research is given by N.M.B. and Z.A.K. The earliest manuscript is verified by Z.A.K. and N.S., Z.A.K. and N.S. have also verified the analytical method and the interpretation of the results of this article. Z.A.K. supervises the revised version of this article as a correspondence author. All authors have contributed significantly from the earlier draft until the final stage of the manuscript. All authors have read and agreed to the published version of the manuscript.

**Funding:** This research received no external funding.

**Conflicts of Interest:** The authors declare no conflict of interest.

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
