# Peer review of "The Effects of Factors of Production Shocks on Labor Productivity: New Evidence Using Panel VAR Analysis"

_sustainability, doi:10.3390/su12208710_

Round 1

Reviewer 1 Report

Introduction

Line 26-31: Those lines are almost the same as in the abstract. The abstract of the paper should summerise (not repeat) its content, therefore I suggest to review those sentences.

Line 112-130: it is used a different font than the recommended one.

Review of Literature

It would be easier to follow this chapter if the reviewed studies would be organized by certain criteria and places under different subtitles.

Methodology and data

Please provide the sources of data series as within-text references and in the final list of references (including the link an access date).

Starting from the methodology section, the within-text reference do not follow the style recommended by MDPI. Moreover (as far as I see) almost all of them are not in the references list. I am sorry, but this is not acceptable for any journal.

Empirical results and discussion

Each table should be referred to in the text before entering it.

Conclusion

Line 564: The beginning of the phrase is missing.

Author Response

Please see the attachment file.

[Many thanks to the Reviewer 1 for the comments/suggestions]

Reviewer 2 Report

Even, if the paper traits a relevant topic, the impact of production shocks on labour productivity using Panel VAR analysis, several observations need to be taken into account:

-the introduction need to be fundamentally rebuild in order to better highlight the role of the paper, the motivation for analyzing all these and how the paper fills the research gap bringing its contribution to the literature.

-although the authors tried to explain each step made to obtain the empirical results, the lack of presentation of the empirical results of the VAR models raises an uncertainty regarding the quality of the research.
That is why I ask the authors to include in the article the table with the results of the models and to adjust the presentation of the empirical results with the inclusion of all these information.

Also, a Granger causality need to be implemented in order to check if the IRF make sense.

-the empirical results and the section of conclusions need to be dramatically re-structured in order to point out in a more clear manner the utility of such a research.

Author Response

[Many thanks to the Reviewer 2 for the comments/suggestions]

Reviewer 3 Report

Dear Authors,

Although the interest of your paper, in order to be published, some improvements are needed, namely:

  • Please insert the main aims of this paper in the abstract section.
  • On page 6, just at the end of fourth paragraph, please delete "3.Results"
  • Authors should insert a "Results Discussion" Section.
  • Authors need to clarify and insert the theoretical and practical contributions of this study to this field of research, in the "conclusion" section.

  •  

    English needs to be improved.

Author Response

[Many thanks to the Reviewer 3 for the comments/suggestions]

Round 2

Reviewer 1 Report

All my suggestions were addressed by the authors.

Author Response

Many thanks to Reviewer 1 for satisfying what we have done in response to the comments/suggestion.

The revised version of the manuscript has been proofread by professional proofreader, and also double checking using Grammarly Premium package. 

Reviewer 2 Report

I would like to thank the authors for their efforts in improving the manuscript.

From my point of view, without testing and further including the results of Granger causality, you risk to under-evaluate your paper using only the propagation of shocks through IRFs.

What is the sense of applying IRFs if there is not a causality between that variables?

It is my major observation and concern. Please take into account and provide some discussions about this topic.

Otherwise, the paper satisfies the requirements for publication.

Author Response

Many thanks to the Reviewer 2 for the comments/suggestion. We have considered all suggestions/comments as requested.

Please refer the attached file.
